# Serum and urinary biomarkers for early detection of acute kidney injury following *Hypnale* spp. envenoming

**Eranga Sanjeewa Wijewickrama**[1,2]*, **Fahim Mohamed**[2,3,4,5], **Indika B. Gawarammana**[2,6], **Zoltan H. Endre**[5], **Nicholas A. Buckley**[2,4], **Geoffrey K. Isbister**[2,7]

**1** Department of Clinical Medicine, Faculty of Medicine, University of Colombo, Colombo, Sri Lanka, **2** South Asian Clinical Toxicology Research Collaboration, Faculty of Medicine, University of Peradeniya, Peradeniya, Sri Lanka, **3** Department of Pharmacy, Faculty of Allied Health Sciences, University of Peradeniya, Peradeniya, Sri Lanka, **4** The University of Sydney, Faculty of Medicine and Health, Biomedical informatics and Digital Health, Clinical Pharmacology and Toxicology Research Group, Sydney, NSW, Australia, **5** Australian Kidney Biomarker Reference Laboratory, Department of Nephrology, Prince of Wales Hospital and Clinical School, University of New South Wales, Sydney, Australia, **6** Department of Medicine, Faculty of Medicine, University of Peradeniya, Peradeniya, Sri Lanka, **7** Clinical Toxicology Research Group, University of Newcastle, Newcastle, New South Wales, Australia

* erangasw@gmail.com

**Data Availability Statement:** All the relevant data are within the manuscript and its Supporting Information files.

## Abstract

### Background

Hump-nosed pit viper (HNV; *Hypnale* spp.) bites account for most venomous snakebites in Sri Lanka. Acute kidney injury (AKI) is the most serious systemic manifestation (1–10%) following HNV envenoming. We aimed to identify the value of functional and injury biomarkers in predicting the development of AKI early following HNV bites.

### Methods

We conducted a prospective cohort study of patients with confirmed HNV envenoming presenting to two large tertiary care hospitals in Sri Lanka. Demographics, bite details, clinical effects, complications and treatment data were collected prospectively. Blood and urine samples were collected from patients for coagulation and renal biomarker assays on admission, at 0-4h, 4-8h, 8-16h and 16-24h post-bite and daily until discharge. Follow-up samples were obtained 1 and 3 months post-discharge. Creatinine (sCr) and Cystatin C (sCysC) were measured in serum and kidney injury molecule-1 (uKIM-1), clusterin (uClu), albumin (uAlb), β2-microglobulin (uβ2M), cystatin C (uCysC), neutrophil gelatinase associated lipocalin (uNGAL), osteopontin (uOPN) and trefoil factor-3 (uTFF-3) were measured in urine. Definite HNV bites were based on serum venom specific enzyme immunoassay. Kidney Disease: Improving Global Outcomes (KDIGO) criteria were used to stage AKI. Two patients had chronic kidney disease at 3 month follow-up, both with pre-existing abnormal sCr, and one developed AKI following HNV envenoming.

**Funding:** GI received National Health and Medical Research Council (NHMRC) Project grant ID: 1011772 and NHMRC Senior Research Fellowship Grant ID: 1154503. NB received Centers of Research Excellence Grant ID: 1110343. The funders played no role in the study design, data collection and analysis, decision to publish, or preparation of the manuscript.

**Competing interests:** The authors have declared that no competing interests exist.

## Results

There were 52 patients with confirmed HNV envenoming; median age 48y (Interquartile range [IQR]:40-59y) and 29 (56%) were male. Median time to admission was 1.87h (IQR:1–2.75h). Twelve patients (23%) developed AKI (AKI stage 1 = 7, AKI stage 2 = 1, AKI stage 3 = 4). Levels of five novel biomarkers, the functional marker serum Cystatin C and the damage markers urinary NGAL, cystatin C, β2-microglobulin and clusterin, were elevated in patients who developed moderate/severe acute kidney injury. sCysC performed the best at 0–4 h post-bite in predicting moderate to severe AKI (AUC-ROC 0.95;95%CI:0.85–1.0) and no biomarker performed better than sCr at later time points.

## Conclusions

sCysC appears to be a better marker than sCr for early prediction of moderate to severe AKI following HNV envenoming.

### Author summary

Snakebite is a major public health problem associated with considerable morbidity and mortality worldwide. Acute kidney injury is one of the major systemic complications of snakebites. Its pathophysiology is poorly understood and the diagnosis is often delayed due to lack of sensitive biomarkers. We aimed to investigate the value of selected biomarkers in the early diagnosis of acute kidney injury following hump-nosed pit viper (*Hypnale* spp.) envenoming. In a group of 52 patients with confirmed hump-nosed pit viper envenoming acute kidney injury was found to be common and was associated with severe disease in some. Levels of five novel biomarkers, the functional marker serum Cystatin C and the damage markers urinary NGAL, cystatin C, β2-microglobulin and clusterin, were elevated in patients who developed moderate/ severe acute kidney injury. Serum Cystatin C performed better than serum creatinine in early prediction of moderate/severe acute kidney injury. Serum Cystatin C appears to be a promising novel biomarker in diagnosing acute kidney injury in the setting of hump-nosed pit viper envenoming.

## Introduction

Snakebite is a major public health problem and snake envenoming is associated with considerable morbidity and mortality worldwide, particularly in Asia and Africa.[1,2] There are five medically important venomous snakes in Sri Lanka, Russell's viper (*Daboia russelii*), common krait (*Bungarus caeruleus*), Indian cobra (*Naja naja*), saw-scaled viper (*Echis carinatus*) and the hump-nosed pit viper (HNV; *Hypnale* spp.). Each results in a characteristic pattern of systemic envenoming, and antivenom is available for the first four of these.

HNV bites account for the majority of venomous snakebites in Sri Lanka, similar to much of India, but severe systemic envenoming occurs in only a small proportion. Envenoming is usually limited to bite site pain and swelling, but more serious uncommon manifestations include coagulopathy, acute kidney injury (AKI), stroke, and myocardial infarction. AKI is the commonest systemic complication and has been associated rarely with death and significant morbidity. The majority of HNV bites therefore require minimal treatment and improved management needs to focus on early identification and treatment of patients that develop AKI.

In addition, HNV bites are thought to be associated with chronic kidney disease (CKD) in Sri Lanka, assumed to be due to prior AKI.[3,4] Identification of AKI following HNV bites may also improve the identification and improve understanding of the development of CKD.

Currently the diagnosis of AKI is based on an acute increase in serum creatinine and a reduction in urine output. Unfortunately, these changes may be delayed and sometime unreliable, [5] and other biomarkers may be more sensitive. It is now recommended by Acute Dialysis Quality Initiative (ADQI) that damage biomarkers be considered as alternatives to functional biomarkers in the diagnosis of AKI.[6] These novel functional and damage biomarkers include urinary and serum cystatin C (uCysC, sCysC), urinary kidney injury molecule 1 (uKIM-1), albumin (uAlb), $\beta_2$ microglobulin (u$\beta$2M), urinary clusterin (uClu), neutrophil gelatinase associated lipocalin (uNGAL), osteopontin (uOPN) and trefoil factor 3 (uTFF-3), which have previously been investigated in poisoned patients and patients with Russell's viper bites.[7,8] These biomarkers may be useful as diagnostic markers of AKI in patients with HNV bites, providing a method of early identification of the small number of patients with severe systemic envenoming.

We investigated the utility of a range of serum and urinary biomarkers for the early identification of AKI following HNV envenoming.

## Materials and methods

### Ethics statement

The study was approved by the Ethics Review Committees of the Faculty of Medicine, University of Peradeniya, and the Faculty of Medicine, University of Colombo (EC-16-109), the University of New South Wales and University of Newcastle, Australia. Formal written consent was obtained from all adult participants. Formal consent was obtained from the parent or the guardian if the participant was a child.

This was a prospective cohort study of patients with definite HNV bites presenting to the Teaching Hospital Peradeniya (April 2016 to September 2018) and the General Hospital Polonnaruwa (April 2012 to March 2015) in Sri Lanka. All patients gave informed written consent for the collection of data, blood samples and urine samples. Formal verbal consent was obtained from the parent/guardian whenever appropriate.

All patients with suspected HNV bites based on a snake specimen, visual description of the snake and/or clinical features, presenting to the two hospitals, were considered for the study. A small number of patients recruited from the Teaching Hospital Peradeniya were also part of a clinical trial conducted to test the efficacy of an experimental polyspecific antivenom, developed against Sri Lankan cobra, Russell's viper, saw-scaled viper and HNV. Patients <15 years, pregnant or with definite evidence of being bitten by snakes other than HNV were excluded. Patients with HNV (*Hypnale* spp.) venom detected in serum by venom specific enzyme immunoassay were included as definite bites. Furthermore, patients were eligible only if they had at least two serum samples and one urine sample available during the first 24 h post-bite.

The following information was collected prospectively: demographic features (age and sex), bite information (snake species and time of bite), clinical effects (local effects, coagulopathy [whole blood clotting time; WBCT] or bleeding, neurotoxicity and myotoxicity), complications, and treatment including antivenom administration. All data were collected on a standard data sheet, previously developed in Sri Lanka for prospective snakebite cohorts.[9,10]

Planned blood (8–10 ml) and urine sample collection times were on admission and within 4 h, between 4 and 8 h, between 8 and 16 h, and between 16 and 24 h post-bite and daily until discharge. After discharge all patients were followed-up at one month and three months at the clinic for both clinical data and serum and urine samples. All samples were immediately

centrifuged and then aliquoted and frozen at -20˚C and then transferred to a -80˚C freezer within 2 weeks of collection.

*Hypnale* spp. venom specific enzyme immunoassay was undertaken as previously described.[11,12] In brief, polyclonal IgG antibodies were raised in rabbits against *H. hypnale* venom antigens. Antibodies were bound to the microplate and also conjugated with biotin as detecting antibodies for a sandwich enzyme immunoassay. The detecting agent was streptavidin–horseradish peroxidase. All samples were assayed in triplicate with triplicate wells having a coefficient of variation of <10% for low and high absorbances. Averaged absorbances were converted to concentrations by comparison with a standard curve. The limit of detection of the assay was 0.2 ng/mL.

Serum creatinine concentrations were measured with the enzymatic method using Clinical and Speciality Chemistry System, Thermo Scientific. Urine creatinine concentrations were measured by the modified Jaffe method (kinetic Jaffe reaction method, rate blank and compensated on Roche Hitachi 912 automatic analyser). Cystatin C was measured in serum using a microparticle-enhanced immune turbidimetric method on a clinical chemistry analyser (Konelab ThermoFisher, Waltham, MA), following the manufacturer's recommendations. uKIM-1 and uClu were quantified using Duo Set enzyme-linked immunosorbent assay (ELISA) kits (R&D systems) following the sandwich ELISA technique according to the instructions of the manufacturer.[8] Intra and inter assay precision for ELISA was <10%. The other urinary biomarkers uAlb, uβ2M, uCysC, uNGAL, uOPN and uTFF-3 were measured using BIO-Plex ProRBM Human Kidney Toxicity Assays panel 2 on the Bio-Plex 200 system (BIO-RAD, USA). Inter and intra assay precisions were <15% and <5% respectively.

AKI was defined according to the Kidney Disease: Improving Global Outcomes (KDIGO) criteria.[13] AKI stage 1 was defined as an increase in serum creatinine by $\geq$ 0.3 mg/dl ($\geq$ 26.5 μmol/l) within 48 h or an increase in serum creatinine to 1.5 to 1.9 times baseline. AKI stage 2 was defined as increase in serum creatinine $\geq$ 2.0 to 2.9 times baseline AKI stage 3 was defined as $\geq$ 3.0 times baseline or increase in serum creatinine to $\geq$ 4.0 mg/dl ($\geq$ 353.65 μmol/l) or initiation of renal replacement therapy. The baseline serum creatinine was defined as the lowest serum creatinine greater than 0.4 mg/dl recorded on admission or during follow up.

Continuous data were reported as medians, interquartile ranges (IQR) and ranges, and categorical data as proportions with 95% confidence intervals. AKI stage 2 and 3 patients were grouped together as moderate to severe AKI and compared to patients with no AKI and AKI stage 1 (mild AKI). The time course of biomarker concentrations was compared between patients with differing severity of AKI. Comparisons and correlations between ordered groups of AKI for venom concentrations and biomarkers were undertaken using Kendall's Tau-b test in SPSS with a one tailed test. The diagnostic performance of each of the biomarkers for HNV associated AKI within each time interval was assessed by area under the receiver operator characteristic curve (AUC-ROC). Normal reference ranges for the serum and urinary biomarker levels were based on the findings of a previous study conducted among healthy adult volunteers.[14] Most statistical analysis was performed using GraphPad Prism software version 8.4 (San Diego, USA).

## Results

From a total of 1,568 patients admitted with snakebites, 209 were suspected HNV bites and consented, and 52 were included with venom assay confirmation and sufficient serum and urine samples (Fig 1).

Twenty-nine of 52 patients (56%) were males and the median age was 48 y (IQR: 40-59y; Range: 15-71y). The median time from bite to admission was 1.87h (IQR: 1 to 2.75h; Range:

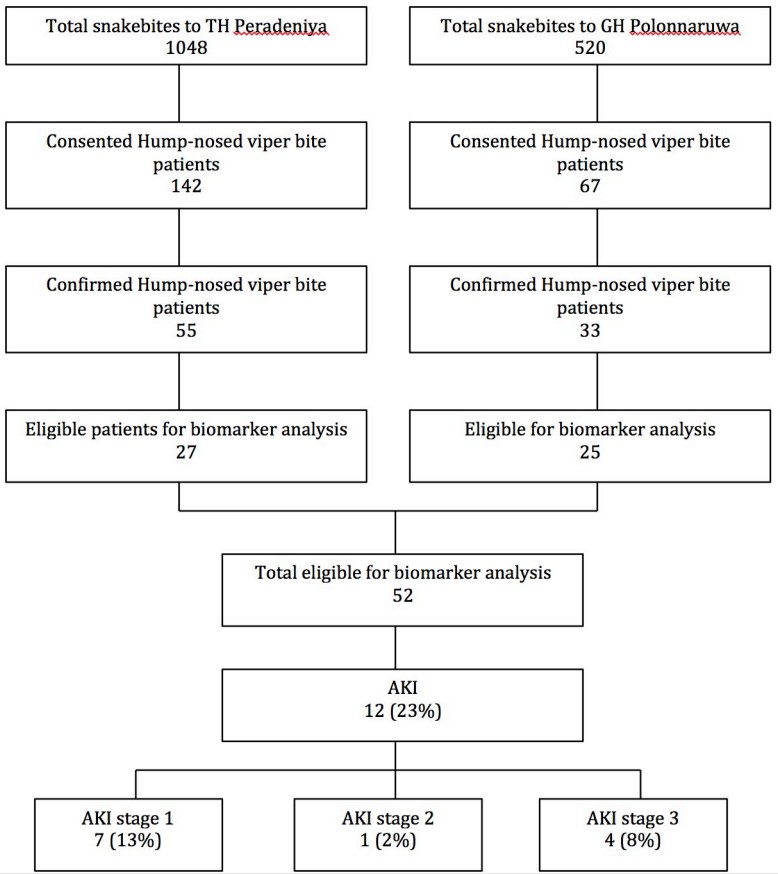

**Fig 1. Flow chart of included patients and grading of acute kidney injury.**

0.5 to 15.3h). Demographic and clinical details of the patients are summarized by stages of AKI in Table 1.

Twelve of 52 patients (23%) developed AKI; 7/52 (13%) AKI stage 1, 1/52 (2%) AKI stage 2 and 4/52 (8%) AKI stage 3. Pain and swelling at the bite site were the commonest clinical manifestations and haematemesis occurred in one patient and bite site bleeding in another. WBCT20 was prolonged in only two patients who did not develop AKI, while it was prolonged in four patients who developed AKI (two stage 1 and two stage 3).

There were no differences between the median venom concentrations in patients with no AKI, mild AKI and moderate/severe AKI (Fig 2). Venom concentrations decreased to the limit of detection rapidly within 12 h of the bite (S1 Fig). Four patients (two with AKI stage 1 and two with AKI stage 2/3) received commercially available antivenom (Vins Bioproduct, Hyderabad and Bharat Serum Institute, India), inappropriately, which is ineffective against *Hypnale* spp. envenoming. Twenty-two patients received the experimental polyspecific antivenom that includes antibodies to HNV venom: 20/38 (53%) with no AKI, 1/6 (17%) with AKI stage 1 and 1/4 (25%) with AKI stage 2/3 (Table 1). The incidence of AKI was 2/22 (9%) in the group that received the experimental antivenom compared to 8/26 (31%) in the group that did not receive this antivenom.

Data points of patients receiving the experimental antivenom are depicted with colour filled circles. Data points of patients not receiving antivenom, receiving commercially available antivenom or information not available on antivenom administration are depicted with clear circles.

**Table 1. Patient demographic and clinical data.**

| | No AKI | AKI stage 1 | AKI stage 2/3 |
|---|---|---|---|
| Number | 40 | 7 | 5 |
| Males (%) | 20 (50) | 7 (100) | 2 (40) |
| Median age (years) (range) | 49 (15–70) | 45 (34–71) | 57 (46–64) |
| Median time to admission (h) (IQR, range) | 1.87 (1.03–2.58, 0.5–5.5) | 1 (1–2, 0.5–4.67) | 2.92 (2.33–9.92, 1.75–15.83) |
| Site of bite[a] | | | |
| Lower limb (n) | 21 | 5 | 4 |
| Upper limb (n) | 18 | 2 | - |
| Local signs and symptoms[a] | | | |
| Pain (%) | 38/39 (97) | 6/7 (86) | 4/4 (100) |
| Swelling (%) | 35/39 (90) | 5/7 (71) | 4/4 (100) |
| Necrosis (%) | 2/39 (5) | - | 4/4 (100) |
| Median serum venom concentration (ng/ml) (range) | 4.1 (0.5–36) | 2.6 (0.5–24) | 3.2 (1.2–46.1) |
| Treatment with antivenom (%)[b,c] - Commercial - Experimental | 20/38 (53) - 20 | 3/6 (50) 2 1 | 3/4 (75) 2 1 |

[a]Data on clinical features were not available for two patients (No AKI = 1, AKI 2–3 = 1)

[b]Data on antivenom administration were not available for four patients (No AKI = 2, AKI 1 = 1, AKI 2–3 = 1)

[c]4/26 patients received commercially available antivenom (Vins Bioproduct, Hyderabad and Bharat Serum Institute, India) due to misidentification of the snake as Russell's viper. 22/26 received the experimental polyspecific antivenom being developed to cover all venomous snakebites in Sri Lanka including HNV bites.

## Peak biomarker concentrations and AKI

The maximum concentration of each biomarker within 24h post-bite for each of the AKI patient group is illustrated in Figs 3 and S2. The correlation of peak biomarker concentration within 24 h post-bite with no AKI, mild AKI, and moderate/severe AKI for each of the renal biomarker is given in S1 Table. Patients with moderate/severe AKI (AKI stage 2/3) had high peak concentrations of sCr, sCysC, uNGAL, uCysC and uβ2M than patients with no or mild AKI. For biomarkers sCr, sCysC, uCysC and uβ2M the peak concentrations within 24 h post-bite in patients with moderate/severe AKI were above the normal reference ranges.

## Time dependent changes in absolute and median biomarker concentrations

Increased concentrations of serum and urinary biomarkers were observed mostly in patients with moderate/severe AKI (AKI stage 2–3) and there were minimal differences in the biomarker levels between patients without AKI and mild AKI (AKI stage 1) (S3 Fig). Similarly, the median concentrations of sCr, sCysC, uNGAL, uCysC, uClu and uβ2M increased in patients with moderate/severe AKI (AKI stage 2/3) over time, but there were only minimal changes in AKI stage 1 compared to no AKI (Fig 4).

Serum creatinine increased over the first 4–8 h post-bite in patients with moderate/severe AKI, with the median sCr continuing to rise at 24–48 h (Fig 4). Serum cystatin C rapidly increased in patients with moderate/severe AKI with median peak sCysC occurring at 16–24 h post-bite (Fig 4). uNGAL rapidly increased by 4–8 h and peaked at 8–16 h post-bite in patients with moderate/severe AKI, then decreased gradually. uCysC increased in patients with moderate/severe AKI from 4–8 h post-bite, peaking at 8–16 h in patients with a prolonged increase until discharge. Median uClu concentration had a delayed rise with a peak at 16–24 h post-bite

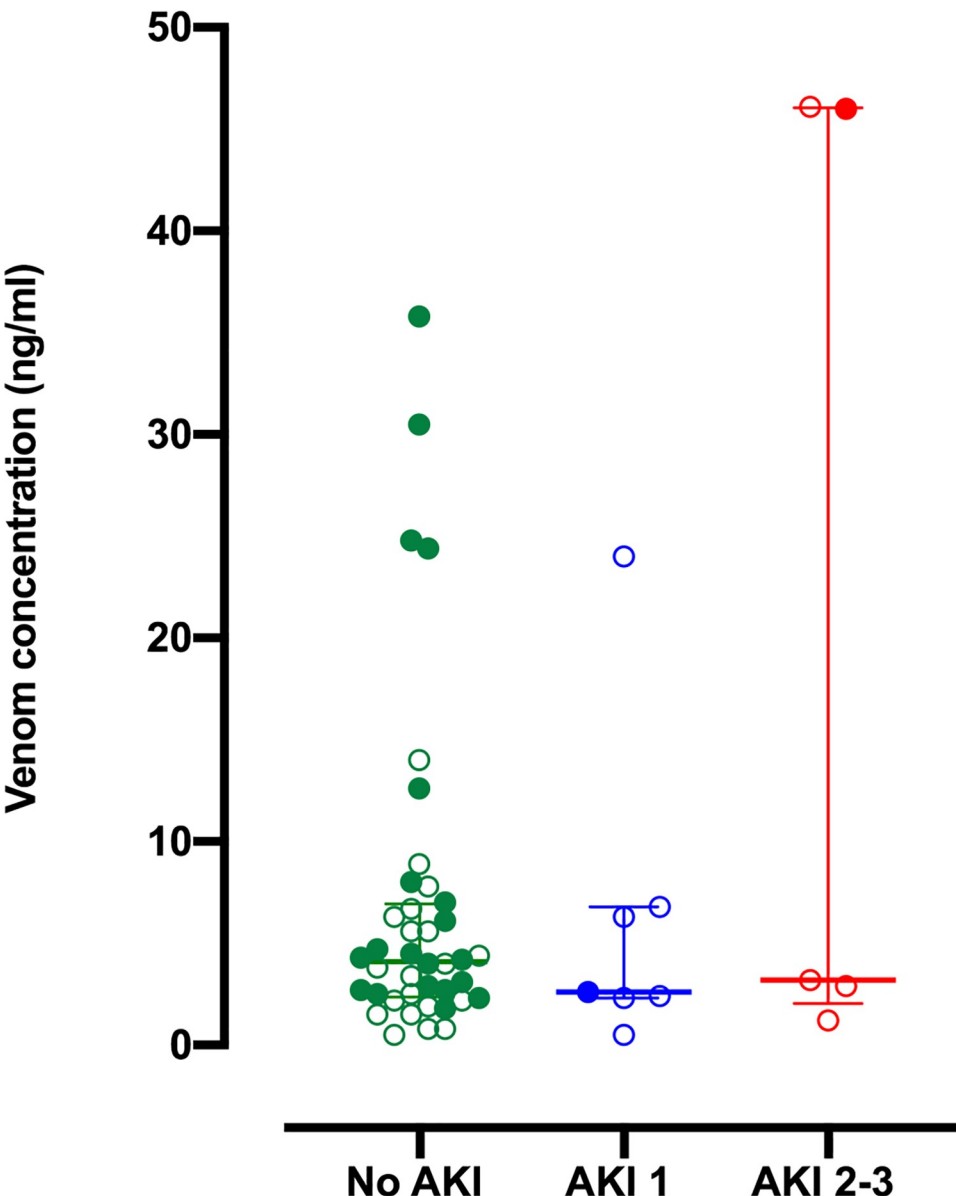

**Fig 2. Scatter plots of the peak HNV venom concentrations in serum within 24 h post bite (median and interquartile range) for each of the acute kidney injury groups.**

in patients with moderate/severe AKI, but was within the reference range at other times. Urinary biomarker uβ2M was elevated on admission in patients with moderate/severe AKI, then decreased before peaking at 24–48 h post-bite.

Concentrations of the other four renal biomarkers, namely uAlb, uTFF3, uKIM-1 and uOPN, either fluctuated within the normal reference range and were not useful in diagnosing AKI or were too variable to distinguish AKI (S4 Fig).

## Predicting moderate/severe AKI versus no AKI/mild AKI

Serum cystatin C was the biomarker with the best performance at 0–4 h post-bite in predicting moderate/severe AKI (AUC-ROC 0.95, 95% CI: 0.85–1.0). A sCysC concentration of >1.34

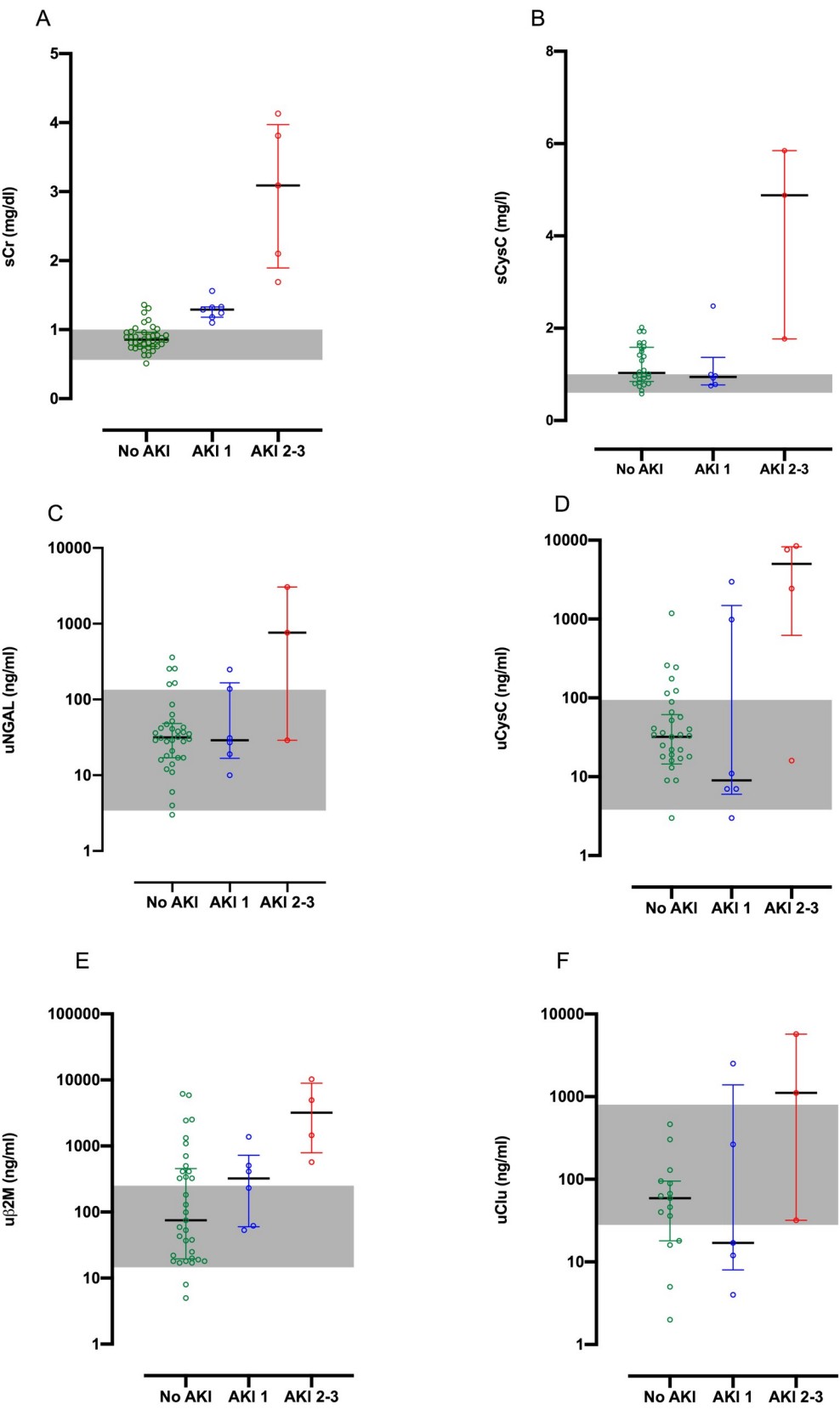

**Fig 3.** Maximum renal biomarker concentration within 24 h of bite. Scatter plots of the maximum biomarker concentrations reached within 24 h post-bite for patients with no acute kidney injury (No AKI; green), mild AKI (AKI 1; blue) and moderate to severe AKI (AKI 2–3; red), for serum creatinine (sCr; Panel A), serum cystatin C (sCysC; Panel B), urinary neutrophil gelatinase associated lipocalin (uNGAL; Panel C), urinary cystatin C (uCysC; Panel D), urinary beta-2 microglobulin (uβ2M; Panel E), and urinary clusterin (uClu; Panel F). The grey shaded is the normal range based on respective biomarkers measured in healthy individuals.

mg/l at 0–4 h post-bite had a sensitivity of 100% and a specificity of 95% in predicting development of moderate to severe AKI. At the same time point a sCr of >0.92 mg/dl and a uNGAL level of >25 ng/ml predicted the development of AKI, each with a sensitivity of 100% but at lower specificity of 69% and 63% respectively. At all other time points serum creatinine performed better or equal to other biomarkers in predicting AKI (Fig 5, Table 2).

## Long-term biomarker levels

sCr concentrations at 1 month and 3 months were available for 31 (60%) patients (No AKI = 23, mild AKI = 4, moderate/severe AKI = 4). Estimated GFR at either 1 month or 3 months was less than 60ml/min/1.73m$^2$ in two patients, one without AKI and one with severe AKI (Fig 6A). One patient had an abnormal creatinine on admission which was persistently elevated at the 3 month follow up, and was considered to have pre-existing CKD without any AKI following the snakebite (Fig 6B). The other patient also had abnormal creatinine on admission but developed further increases after admission qualifying for diagnosis of AKI in addition to pre-existing CKD (Fig 6B).

## Discussion

In patients with definite HNV envenoming a number of renal biomarkers increased in addition to sCr, the traditional biomarker of AKI. These included sCysC, uNGAL, uCysC, uβ2M and uClu, although the time course and degree of elevation differed for each biomarker. Increased biomarker concentrations were mainly seen in patients with moderate/severe AKI, while there was less difference in biomarkers between patients who did not develop AKI and mild AKI. sCysC predicted moderate/severe AKI better than sCr in a small window from 0 to 4 h post-bite, but after this time no other biomarker performed better than sCr. uAlb, uTFF-3, uKIM-1 and uOPN performed poorly at any time after the bite compared to sCr.

Serum and urinary cystatin C have been shown to be highly predictive of AKI in a variety of clinical settings associated with exposure to nephrotoxic substances. These include glyphosate surfactant poisoning [15], contrast induced nephropathy [16,17] and nephrotoxicity from drugs such as cisplatin [18] and colistin.[19] sCysC had a modest diagnostic performance (AUC-ROC between 0.65–0.8) in predicting moderate to severe AKI at 0–8 h post-bite following Russell's viper envenoming, which was better than sCr.[8] Also in this study sCysC showed excellent performance in predicting severe AKI early (0–4 h post-bite) following HNV envenoming and performed better or equal to sCr at all times within 24 h in predicting the development of moderate/severe AKI. Furthermore, sCysC concentration is only minimally affected by sex, age, height and muscle mass, so it is an appropriate renal biomarker in children, elderly and individuals with less muscle mass.

Four other biomarkers were indicative of moderate/severe AKI in HNV envenoming, but were no better than sCr. uNGAL appeared to be the best of these, with an increasing AUC-ROC over 24 h, but never better than sCr or sCysC. This differs to a recent study of Russell's viper envenoming in Sri Lanka, in which uNGAL concentrations were elevated in all envenomed patients, irrespective of the AKI status.[8] Although elevated uNGAL was an

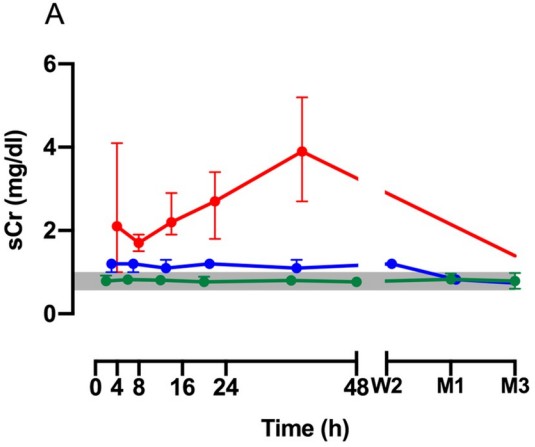

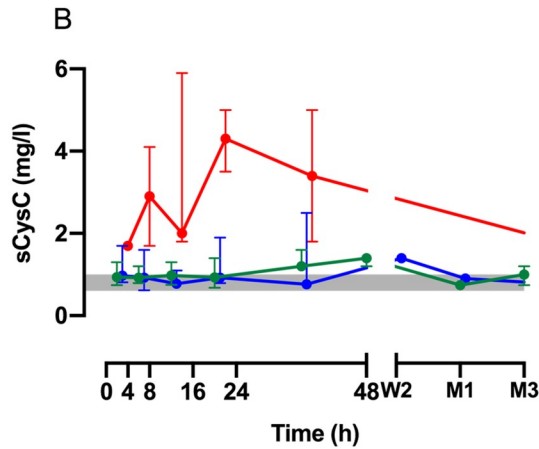

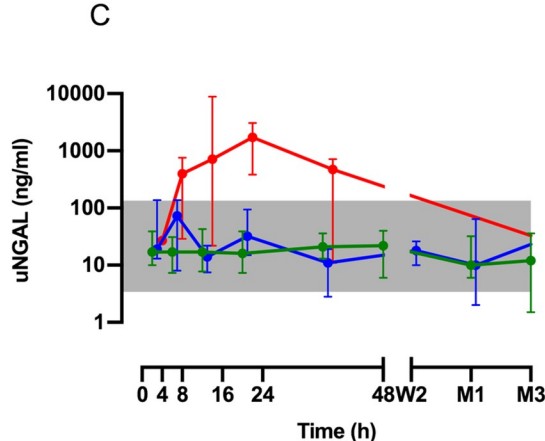

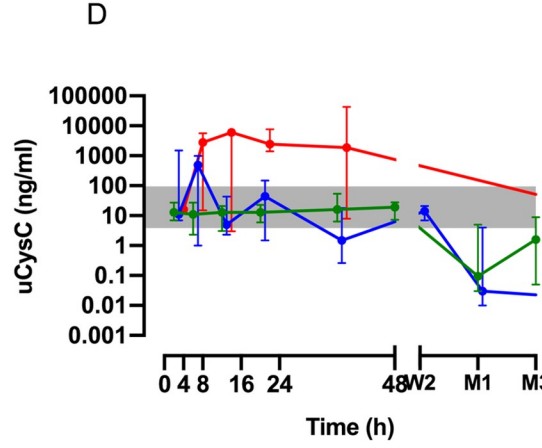

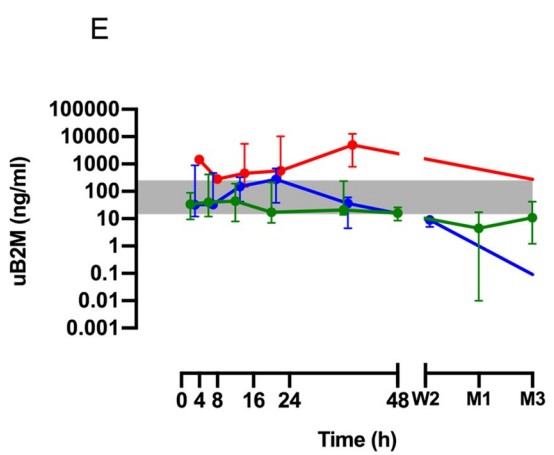

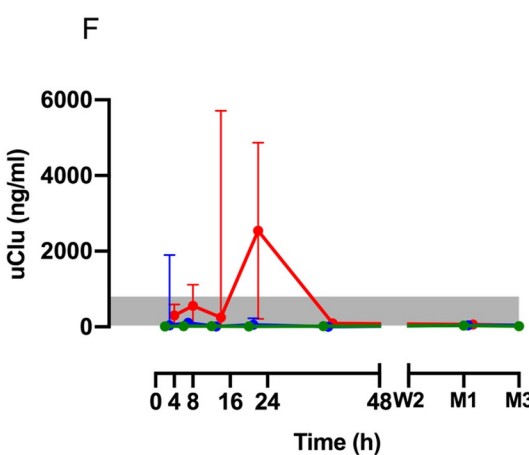

**Fig 4.** Time course of the median biomarker concentrations (with interquartile ranges) for each of the three patient groups ([No AKI; green], [Mild AKI; blue] and [moderate to severe AKI; red]) post-bite for 48 h including serum creatinine (sCr; Panel A), serum cystatin C (sCysC; Panel B), urinary neutrophil gelatinase associated lipocalin (uNGAL; Panel C), urinary cystatin C (uCysC; Panel D), urinary beta-2 microglobulin (uβ2M; Panel E), and urinary clusterin (uClu; Panel F). The grey shaded is the normal range based on respective biomarkers measured in healthy individuals.

excellent marker of Russell's viper envenoming, the extent of uNGAL elevation showed only a modest performance in predicting the development of moderate/severe AKI (AUC-ROC values of 0.64, 0.74, 0.79 and 0.75 at 0–4 h, 4–8 h, 8–16 h and 16–24 h post-bite respectively). This contrasted with a study of Russell's viper bites in Tamil Nadu, where a plasma NGAL level of >245ng/dl on admission was highly predictive of AKI compared to serum creatinine.[20] However, that study did not use accepted standard criteria in diagnosing AKI and there was little information on time of admission post-bite, snake species or snake identification. A similar study found that at the time of admission plasma NGAL was a better predictor of AKI (diagnosed based on RIFLE criteria) than serum creatinine.[21] In *Bothrops* snakebites presenting to an emergency centre in eastern Brazil [22], uNGAL levels were significantly higher in patients who developed AKI, but again were inferior to sCr (AUC-ROC of 0.75 versus 0.86, respectively).

Among the other biomarkers studied, uCysC, uβ2M and uClu showed modest performance at least at some time points post-bite in predicting development of severe AKI. However, they were inferior to sCr, sCysC and uNGAL and therefore unlikely to be useful in predicting severe AKI in this setting. The levels of uAlb, uTFF3, uOPN and uKIM-1 were generally low and either within the normal reference range or poor predictors of development of severe AKI.

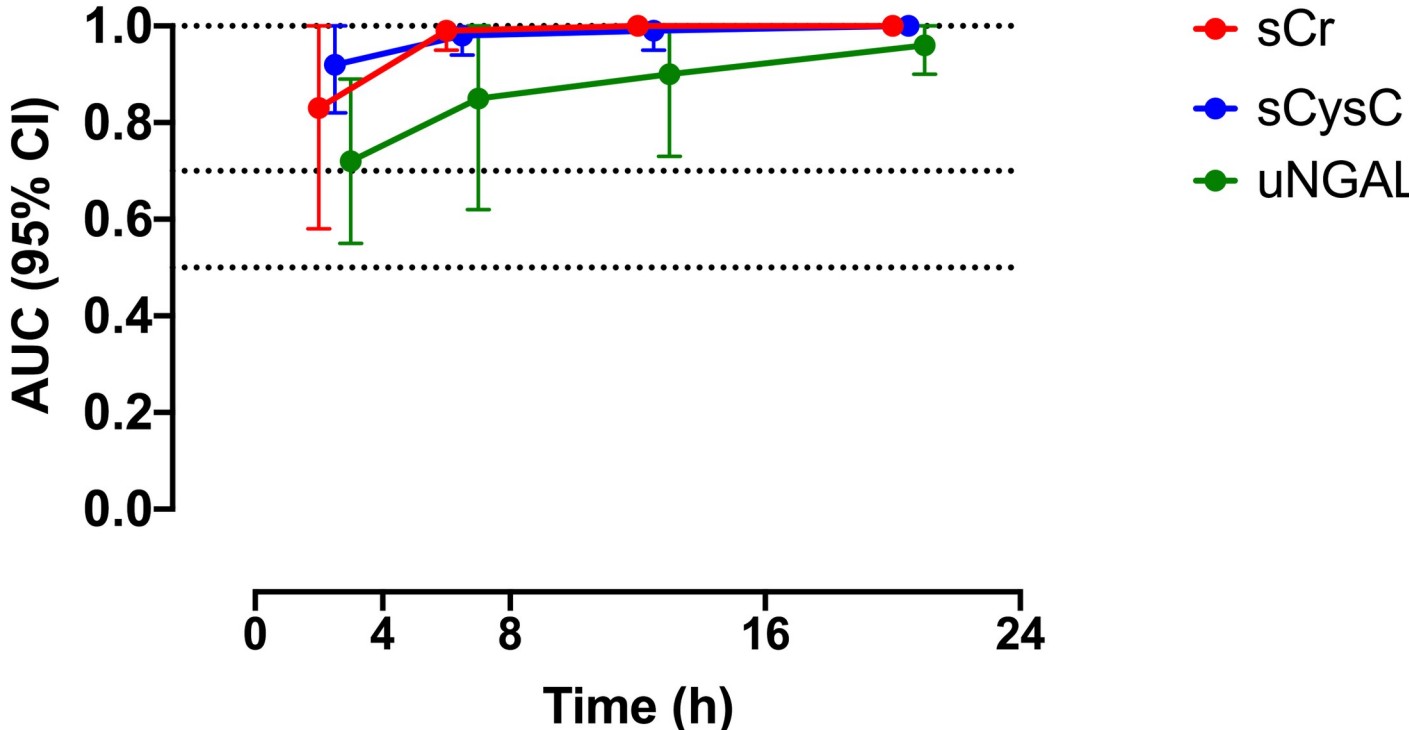

**Fig 5.** Plots of the AUC-ROCs versus time for the best biomarkers in detecting moderate/severe AKI versus no AKI/mild AKI, including serum creatinine (sCr; red line), serum cystatin C (sCysC; blue line), and urinary neutrophil gelatinase-associated lipocalin (uNGAL; green line).

**Table 2. The area under the curve of the receiver operator characteristic (AUC-ROC) curve (with 95% confidence intervals) for biomarker concentrations predicting moderate/severe AKI (AKI stage 2–3) versus none or mild AKI (AKI stage 1), in each of the four time periods within the first 24h of the bite.**

|        | AUC-ROC (95% CI) | | | |
|--------|------------------|------------------|------------------|------------------|
|        | 0-4h             | 4-8h             | 8-16h            | 16-24h           |
| sCr    | 0.90 (0.72–1.0)  | **0.99 (0.96–1.0)** | **1.0**       | **1.0**          |
| sCysC  | **0.95 (0.85–1.0)** | **0.98 (0.91–1.0)** | **0.98 (0.93–1.0)** | **1.0**  |
| uNGAL  | 0.67 (0.49–0.84) | 0.85 (0.64–1.0)  | **0.87 (0.65–1.0)** | **1.0**       |
| uCysC  | 0.55 (0.37–0.73) | 0.81 (0.53–1.0)  | 0.75 (0.35–1.0)  | **1.0**          |
| uβ2M   | 1.0*             | 0.66 (0.49–0.84) | 0.84 (0.69–1.0)  | 0.90 (0.74–1.0)  |
| uClu   | 0.66 (0.23–1.0)  | 0.56 (0–1.0)     | 0.83 (0.58–1.0)  | **0.97 (0.88–1.0)***  |
| uAlb   | 0.72 (0.56–0.89) | 0.60 (0.08–1.0)  | 0.63 (0.14–1.0)  | **0.90 (0.80–1.0)** |
| uTFF3  | 0.80 (0.66–0.94) | 0.55 (0.37–0.72) | 0.75 (0.57–0.94) | 0.81 (0.58–1.0)  |
| uKIM-1 | 0.60 (0.27–0.93) | 0.57 (0.08–1.0)  | 0.60 (0.38–0.82) | 0.52 (0.23–0.80) |
| uOPN   | 0.81 (0.67–0.94) | 0.52 (0.17–0.86) | 0.64 (0.25–1.0)  | 0.85 (0.69–1.0)  |

*Only 1 value for AKI 3

Urinary TFF-3 was shown to be of value in early prediction of AKI from drug-induced nephrotoxicity due to gentamicin [23] and cisplatin [24], and following cardiac arrest.[25] However, similar to HNV envenoming it was found to be a poor predictor of development of moderate/severe AKI following Russell's viper envenoming.[8]

Sri Lankan *Hypnale* spp. venom consists of a number of toxic proteins and peptides including phospholipase A2 (PLA$_2$), snake venom metalloproteinase (SVMP), snake venom serine proteinase (SVSP), L-amino acid oxidase (LAAO) and C-type lectin (CTL).[26] The common local effects are thought to be due to SVMPs, PLA$_2$ and LAAO in venom, which are known to have cytolytic and haemotoxic properties. The coagulopathy is mainly attributed to SVSPs, which have thrombin-like enzyme (TLE) activity. The exact mechanism and toxins responsible for development of AKI in *Hypnale* spp. envenoming are not known. Direct venom nephrotoxicity, venom-induced hemodynamic disturbances, haemolysis, and rhabdomyolysis are some of the mechanisms, which have been postulated.[7] The high levels of uNGAL, uCysC and uβ2M detected in patients with moderate to severe AKI in the present study suggest renal tubular injury in *Hypnale* spp. envenoming. Interestingly, *Hypnale* spp. venom concentrations in plasma were not correlated with the development and severity of AKI in the present study. This could be a result of the wide variability observed in the time to presentation post-bite (S2 Fig). Furthermore, there is a possibility that venom-concentration independent mechanisms operate in AKI following *Hypnale* spp. envenoming. Further studies are required to determine whether AKI is caused by direct venom toxicity or by the indirect effects of the venom, such as thrombotic microangiopathy.

There are a number of limitations to our study. Firstly the diagnosis and staging of AKI was based on KDIGO criteria, which relies on a change in sCr, and not on histopathology, which is the gold standard in diagnosing AKI. However, performing renal biopsies to evaluate kidney injury is neither justifiable nor practical in the setting of snakebite. Another issue was the lack of a baseline sCr, which is required in order to assess the rise of change in sCr for the KDIGO criteria. In the absence of a baseline sCr we used the lowest value of measured serum creatinine, usually at admission or at follow-up.

A possible limitation of our study was the use of two different antivenoms in some patients and not in others. Currently available commercial antivenom (Vins Bioproduct, Hyderabad and Bharat Serum Institute, India) is not effective against HNV envenoming, but was given to four patients, all with AKI. In addition 22 patients recruited from TH Peradeniya were part of

**A**

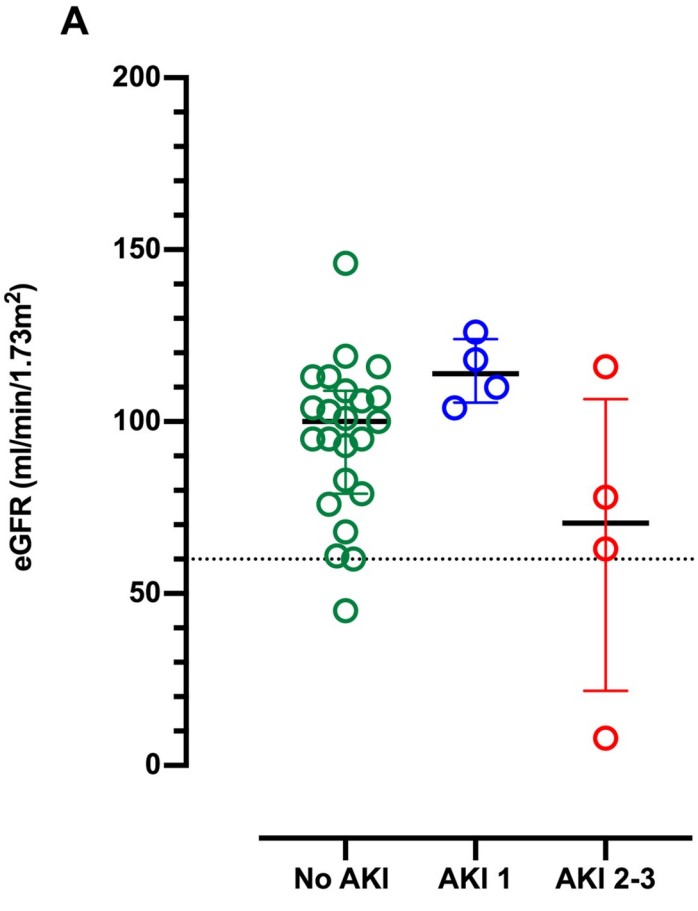

**B**

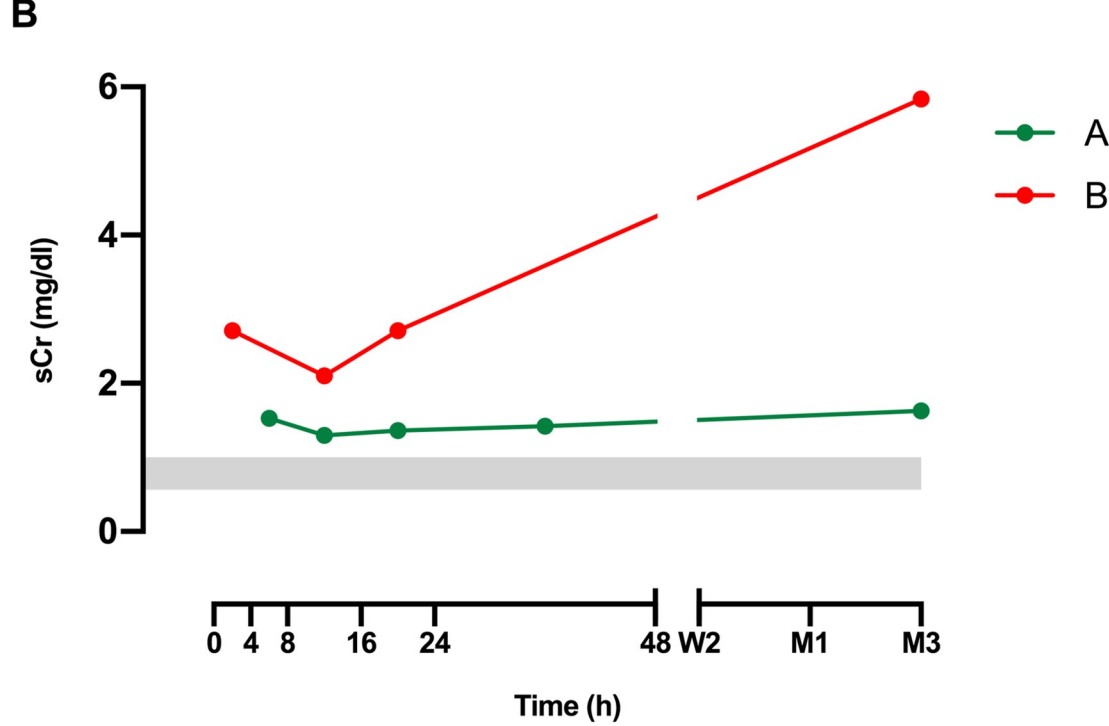

**Fig 6.   A.** Scatter plots of the estimated GFR at 1 month or 3 months* in patients with no acute kidney injury (No AKI; green), mild AKI (AKI 1; blue) and moderate/severe AKI (AKI 2–3; red). The dotted line indicates the normal eGFR of 60ml/min/1.73m². **B.** Time course of sCr concentrations for the two patients with CKD. (Patient A; green line, patient B; red line) The grey shaded is the normal range for sCr measured in healthy individuals. *In all patients eGFR at 3 months was considered if available. In the absence of an eGFR at 3 months eGFR at 1 month was considered.

another clinical trial to evaluate the efficacy of experimental polyspecific antivenom against Sri Lankan cobra, Russell's viper, saw-scaled viper and HNV.[27] The incidence of AKI was lower (9%) in the group that received the experimental antivenom compared to the group that did not receive this antivenom (31%). Therefore, it is possible that the experimental antivenom may have reduced the proportion of patients with AKI and the severity of AKI, but the sample size was too small to confirm this.

Another limitation was that not all structural or functional biomarkers of AKI were assessed in our study. Several other biomarkers have been shown to be useful in recognizing AKI in snakebite, including urinary monocyte chemotactic protein-1 (MCP-1) in Bothrops [22] and urinary N-acetyl-β-D-glucosaminidase (NAG) in Russell's viper envenoming.[28] Finally, the number of patients included in the study was only 52 and just seven and five patients had mild and moderate to severe AKI, respectively.

Although we provide some support for novel renal biomarkers being useful in HNV envenoming, the clinical utility of these biomarkers is dependent on the availability of rapid testing and specifically point-of-care assays. Currently there are immunoassay based point-of-care devices and platforms for measurement of NGAL in blood and urine with the ability to provide results in less than one hour.[29] Similarly a point-of-care test based on latex enhanced immunoturbidimetric assay is available for serum cystatin C (EUROLyser), which can provide results within 10 minutes (Eurolyser Diagnostica GmbH, Salzburg, Austria).[30] The increased availability of these point-of-care assays will be helpful in identifying individuals at risk of developing AKI early so that specific interventions could be implemented in a timely manner. This is especially relevant to HNV bites in which AKI is by far the most significant manifestation of systemic envenoming, but it only occurs in a small proportion of envenomed patients. A specific intervention to prevent/treat AKI in the setting of HNV envenoming is currently not available, but this is likely to change soon with the development of the new polyspecific antivenom currently undergoing clinical trials.[27]

Severe AKI following Russell's viper and HNV envenoming has been associated with the development of CKD.[3,4] Identification of individuals who are likely to develop CKD following AKI due to envenoming is important in deciding on the long-term management and follow-up of these patients. The serum and urinary levels of several studied biomarkers–sCr, sCysC, uβ2M, uClu, uAlb, uCysC, and uNGAL—were elevated in the immediate post-bite period in patients who went on to develop CKD. This suggests the potential utility of these biomarkers in predicting the development of CKD in the setting of HNV envenoming and the need for further studies to explore the validity of this observation.

In conclusion, we found that serum cystatin C level within 4 h of the bite was the best predictor of developing severe AKI following HNV envenoming. At all other time points serum creatinine was the best predictor in development of severe AKI. Further studies are required to evaluate the precise mechanism of venom induced kidney injury and look specifically how sCysC and uNGAL relate to this mechanism in predicting development of snakebite associated AKI.

## Supporting information

**S1 Table. Peak biomarker concentration reached within 24h post-bite. Absolute biomarker concentrations are presented as medians and interquartile ranges (No AKI, AKI stage 1**

**and AKI stage 2–3 were compared and correlated using Kendall's tau-b test with single tailed p-value).** *Correlation coefficients and p values given for No AKI vs AKI 2–3.
(DOCX)

**S1 Fig. Venom concentrations plotted against time from the bite for patients with HNV envenoming for each of the acute kidney injury groups.**
(TIF)

**S2 Fig.** Maximum renal biomarker concentration within 24 h of bite. Scatter plots of the maximum biomarker concentrations reached within 24 h post-bite for patients with no acute kidney injury (No AKI; green), mild AKI (AKI 1; blue) and moderate to severe AKI (AKI 2–3; red), for urinary albumin (uAlb; Panel A), urinary trefoil factor-3 (uTFF3; Panel B), urinary kidney injury molecule-1 (uKIM-1; Panel C) and urinary osteopontin (uOPN; Panel D). The grey shaded is the normal range based on respective biomarkers measured in healthy individuals.
(TIF)

**S3 Fig. Absolute changes in the biomarker concentrations following HNV bite for each patient over the first three months.** Patients without AKI ([No AKI; green], with AKI stage 1 [Mild AKI; blue] and with AKI stages 2–3 [moderate/severe AKI; red]). serum creatinine (sCr; Panel A), serum cystatin C (sCysC; Panel B), urinary neutrophil gelatinase associated lipocalin (uNGAL; Panel C), urinary cystatin C (uCysC; Panel D), urinary beta-2 microglobulin (uβ2M; Panel E), urinary clusterin (uClu; Panel F), urinary albumin (uAlb; Panel G), urinary trefoil factor-3 (uTFF3; Panel H), urinary kidney injury molecule-1 (uKIM-1; Panel I) and urinary osteopontin (uOPN; Panel J). The grey shaded is the normal range based on respective biomarkers measured in healthy individuals.
(TIF)

**S4 Fig.** Time course of the median biomarker concentrations (with interquartile ranges) for each of the three patient groups ([No AKI; green], [Mild AKI; blue] and [moderate to severe AKI; red]) post-bite for 48 h including urinary albumin (uAlb; Panel A), urinary trefoil factor-3 (uTFF3; Panel B), urinary kidney injury molecule-1 (uKIM-1; Panel C) and urinary osteopontin (uOPN; Panel D). The grey shaded is the normal range based on respective biomarkers measured in healthy individuals.
(TIF)

## Acknowledgments

We acknowledge the assistance given by the medical and nursing staff of General Hospital, Polonnaruwa and Teaching Hospital, Peradeniya and the staff of South Asian Clinical Toxicology Research Collaboration (SACTRC) for their support.

## Author Contributions

**Conceptualization:** Fahim Mohamed, Indika B. Gawarammana, Zoltan H. Endre, Nicholas A. Buckley, Geoffrey K. Isbister.

**Data curation:** Eranga Sanjeewa Wijewickrama, Fahim Mohamed.

**Formal analysis:** Eranga Sanjeewa Wijewickrama, Fahim Mohamed, Nicholas A. Buckley, Geoffrey K. Isbister.

**Funding acquisition:** Nicholas A. Buckley, Geoffrey K. Isbister.

**Investigation:** Fahim Mohamed, Indika B. Gawarammana.

**Methodology:** Nicholas A. Buckley, Geoffrey K. Isbister.

**Project administration:** Fahim Mohamed, Indika B. Gawarammana.

**Resources:** Nicholas A. Buckley, Geoffrey K. Isbister.

**Supervision:** Indika B. Gawarammana, Zoltan H. Endre, Nicholas A. Buckley, Geoffrey K. Isbister.

**Writing – original draft:** Eranga Sanjeewa Wijewickrama.

**Writing – review & editing:** Fahim Mohamed, Indika B. Gawarammana, Zoltan H. Endre, Nicholas A. Buckley, Geoffrey K. Isbister.

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
