## [Decision Letter · Decision Letter 0]

8 Sep 2021

Dear Prof. Wijewickrama,

Thank you very much for submitting your manuscript "Serum and urinary biomarkers for early detection of acute kidney injury following Hypnale spp. envenoming" for consideration at PLOS Neglected Tropical Diseases. As with all papers reviewed by the journal, your manuscript was reviewed by members of the editorial board and by several independent reviewers. In light of the reviews (below this email), we would like to invite the resubmission of a significantly-revised version that takes into account the reviewers' comments. 

The manuscript was evaluated by three expert reviewers that acknowledged the relevance of the study for prognosis and treatment of Hump-nosed envenomings in Sri Lanka. However, important points were addressed by the reviewers, indicating that a major revision is fundamental before the publication of the manuscript. The major point of consideration was the sample size in AKI1 and AKI2-3 group that may not be sufficient to prove the hypothesis. Moreover, some patients also participate of another study that evaluates a polyspecific antivenom in a clinical trial and eventual interference of the antivenom in the AKI severity evaluations needs to be considered as an independent group. The other issues addressed are also very important and should be considered in the revision of the manuscript.

We cannot make any decision about publication until we have seen the revised manuscript and your response to the reviewers' comments. Your revised manuscript is also likely to be sent to reviewers for further evaluation.

Sincerely,

Ana M. Moura-da-Silva

Guest Editor

Wuelton Monteiro

Deputy Editor

The manuscript was evaluated by three expert reviewers that acknowledged the relevance of the study for prognosis and treatment of Hump-nosed envenomings in Sri Lanka. However, important points were addressed by the reviewers, indicating that a major revision is fundamental before the publication of the manuscript. The major point of consideration was the sample size in AKI1 and AKI2-3 group that may not be sufficient to prove the hypothesis. Moreover, some patients also participate of another study that evaluates a polyspecific antivenom in a clinical trial and eventual interference of the antivenom in the AKI severity evaluations needs to be considered as an independent group. The other issues addressed are also very important and should be considered in the revision of the manuscript.

Reviewer's Responses to Questions

**Key Review Criteria Required for Acceptance?**

**Methods**

-Are the objectives of the study clearly articulated with a clear testable hypothesis stated?

-Is the study design appropriate to address the stated objectives?

-Is the population clearly described and appropriate for the hypothesis being tested?

-Is the sample size sufficient to ensure adequate power to address the hypothesis being tested?

-Were correct statistical analysis used to support conclusions?

-Are there concerns about ethical or regulatory requirements being met?

Reviewer #1: Objective and hypothesis are clear

Sample size seems to be reasonable for this type of outcome

Authors mentioned ethical consent for patients enrolled in study

For other notes and comments see attached letter.

Reviewer #2: 1. Sample size is not sufficient for prove the hypothesis in AKI1 and AKI2-3 group.

Reviewer #3: The methods used in the present work are well described and performed properly to test the hypothesis stated. Ethical aspects were also adequate. 

- The overall population size of the sutudy was very impressive, and the number of elegible patients for the study comprehensive. However, the most critical point of the work is the sample size of the AKI 2-3 group, composed of only 5 patients. With this population size, the authors were not able to perform any statistical significance between groups, which makes the results only suggestive and not conclusive. As a suggestion, the authors could perform the study of AKI predictive markers without classifying the event's severity (AKI patients and No AKI patients). Accepting this, the authors would have enough and conclusive data. In this case, sCysC values are found to be elevated in both AKI1 and AKI2-3 patients, where it is very likely that the prediction results for this biomarker are valid and relevant for AKI (regardless of the severity of the event).

Also, in this manner, the authors could also evaluate the difference of the development of AKI in patients submitted to the two possible antivenoms used in for the treatment. 

- For the AKI classification, the authors use a baseline value for the markers (serum or urine) “defined as the lowest serum creatinine greater than 0.4 mg/dl recorded on admission or during follow up”. It is also known that AKI cases are late events, manifesting in most cases after 48 hours. However, once the data "time to assistance" were clear to all patients (only results in table 1), there would be a possibility of a patient being admitted 24 hours after the accident (common in several regions of the world) and already having a picture of renal dysfunction being installed. Thus, baseline-based AKI classification could be compromised. Do the authors limit the length of "time to assistance" in patients' eligibility criteria? It might be interesting for the authors to inform what was the time limit for treating patients in the article.

- Still in the METHODS section, the authors mention “Normal reference ranges for the serum and urinary biomarker levels were based on the findings of a previous study conducted among healthy adult volunteers.(14)". The reference is based on a previous study, where most of the authors are the same as the present study. However, these are samples quantified at different times and perhaps using different batch. As many kits are not routinely used in the laboratory, the variation between dosages can be large. Also, the difference in population group may be a biased factor. In summary, do the authors consider that this can interfere with the values determined as “normal reference ranges”?

**Results**

-Does the analysis presented match the analysis plan?

-Are the results clearly and completely presented?

-Are the figures (Tables, Images) of sufficient quality for clarity?

Reviewer #1: Results and data analysis seems to be clear, but I have some specific comments/suggestions that are detailed in attached letter.

Reviewer #2: Moreover, the population size in AKI stage 1 and 2-3 is too small (n= 7 and n = 5) and should not be a proper representative population. Also, the biomarker scatter plot such as uCysC and uClu in AKI 1 as well as sCysC and uClu in AKI 2-3 is considerably large indicating that the patient factor might interfere the biomarker level. The author should increase the sample size in AKI group or should take individual factors into account to reduce the error of the analysis.

Suggestion: In case the sample size cannot be increased, other statistical analysis such as linear regression should be analyzed.

Reviewer #3: The results are solid and have an excellent number of markers assessed, all well suited and directly related to AKI events.

The authors noted that Cystatin C was an important biomarker in predicting a severe/moderate event within 4 hours of poisoning, giving credibility to its use in the clinical approach.

- In the RESULTS section "Peak biomarker concentrations and AKI" in Figure 3. The authors should also mention that the increased biomarkers values of moderate/servere AKI patients were also above “normal reference range” (healthy volunteers ), indicating a trust applicability in using the mentioned markers as predictors. Also, biomarkers sCr and 2β2M were also increased in AKI I patients (compared to healthy donor), indicating that both are also valuable markers to predict AKI independently if mild, moderate or severe. This is also valid for others results. 

- Still in the same section, the authors use correlation using Kendall’s tau-b test. However, in S1 table, is not possible to determine to which comparisons the correlation coefficient and p value are (No AKI vs AKI 1 or No AKI vs AKI 2-3, AKI1 vs AKI 2-3??).

- In the RESULTS section" Long-term biomarker levels", the authors could explore better the results, showing that AKI2-3 patients presented decreased eGFR compared to others, but still (the median) within normal limits. Also, the authors focus on two patients (1 from AKI2-3 group and other from No AKI) who presented decreased levels of eGFR (CKD) and showed long term sCr values (Figure 6B). In Figure 6B, patient A (blue) was from AKI2-3 group and patient B (red) from No AKI group?

**Conclusions**

-Are the conclusions supported by the data presented?

-Are the limitations of analysis clearly described?

-Do the authors discuss how these data can be helpful to advance our understanding of the topic under study?

-Is public health relevance addressed?

Reviewer #1: -Are the conclusions supported by the data presented?

In some aspects, yes (for details see comments/suggestion in the letter attached to this review)

-Are the limitations of analysis clearly described? In some aspects, yes

-Do the authors discuss how these data can be helpful to advance our understanding of the topic under study? Some points deserves more attention to discuss, mainly the mechanistic part involved in kidney injury

-Is public health relevance addressed? Yes, the study has a valuable epidemiological significance mainly addressing a local problem of envenomation in Sri Lanka.

Reviewer #2: This manuscript proposed to investigate the novel biomarkers for kidney injury detection in patients with HNV envenoming. The biomarkers which were chosen in this study have been thoroughly studied in previous reports despite the differences of the toxicants. 

The outstanding biomarkers in patients with snake envenoming in this manuscript are not different from those demonstrated in patients with other toxicants. 

Suggestion: Further study should be focused and discovered the precise mechanism of venom that induce kidney injury and look specifically for the novel biomarkers responding to that mechanism. Cystatin C have been well acknowledged as a best for AKI prediction for several nephrotoxic substances so it not surprises when in HNV envenoming patients with AKI had high level of Cystatin C. This indicates that Cystatin C is not a specific markers for HNV envenoming.

Reviewer #3: The conclusion is well prepared and adequately discusses the results and conclusions obtained. It also shows the benefits of the results in improving the clinical management of the patient.

**Editorial and Data Presentation Modifications?**

Reviewer #1: Minor points:

1. Please review table 1 for layout and format errors. Information of median time to admission for “AKI stage 2/3” column is incomprehensible and in different format comparing to other columns.

2. Figure 2 presents the same information about venom concentration previously showed in table 1? It was measured in serum or urine samples? Please include this information both in table 1 and figure 2 captions.

3. Reference citations seems to be out of journal format. Please double check journal rules.

Reviewer #2: (No Response)

Reviewer #3: - Table is misconfigured.

- Y axis of Figure 3E, correct 2β2M

- In the DISCUSSION section, the authors should mention the sampling limitation concerning the N of patients with AKI2-3

**Summary and General Comments**

Reviewer #1: Comments to the author:

This study presents a clinical evaluation of several classical kidney injury biomarkers for the detection of acute kidney injury (AKI) severity in patients envenomed by the hump-nosed pit viper (Hypnale spp). A range of ten different urinary and serum kidney injury biomarkers were measured in different time-points including in follow up of two weeks, one and three months-post envenomation. A reasonable sample size was included for this type of outcome. Certainly, it has valuable epidemiological significance mainly addressing a local problem of envenomation in Sri Lanka.

Bellow is described major and minor points that the authors should consider for review.

Major points:

1. The major weakness of this study seems to be the fact that 23 of 52 (44%) of patients eligible and included for biomarker tests were also participants of other study that evaluates a polyspecific antivenom in a clinical trial. Thus, it is possible that some of these patients had received antivenom therapy and probably this event has some interference in AKI severity evaluations, mainly during the later follow-up phases of study. The authors only mention this fact in discussion section (page 33). I think this fact must be included and detailed in methods section. The data analysis for this group of patients also needs to be considered as an independent group.

2. Little is mentioned by the authors about venom composition, its mechanisms of action, possible mechanism of renal injury etc… I think it would be interesting to include more information about that. 

3. Since the authors have access to clinical samples (serum and urine) in different time-points for the different severity groups why not include also a panel of serum/urine biochemistry and hematological data?

4. Since heme released from hemoglobin/myoglobin is an important event to be considered in venom-induced nephrotoxicity caused by snakes, did the authors check if patients have myo/hemoglobinuria? How about hemoglobin serum levels? Hemolysis is a common event in Hypnale spp envenomation?

5. How about the imaging exams from these patients? This kind of data also would be interesting to include, since the authors probably have access to clinical data.

6. The median venom serum concentration detected in envenomed victims seems not correlate with AKI severity. How did the authors explain this data? How about the venom clearance by the kidney? Urinary venom concentration was measured? There is some direct nephrotoxin with action on kidney structure already identified in HNV?

7. Patients that evolved to AKI had concomitantly coagulation disturbances? It would be interesting to include some discussion about that. Triggering coagulation and contact systems can cause different changes in renal physiology, like intraglomerular thrombus deposition, changes in glomerular vessel permeability, reduction in GFR, electrolyte tubular transport and changes in urinary concentration ability by collecting ducts.

8. How the authors explain the best performance of serum creatinine as a kidney injury biomarker when compared to urinary NGAL, which is a specific protein released by renal tubules upon injury? A more critical discussion and accurate data literature comparison about this result it would be interesting.

Reviewer #2: (No Response)

Reviewer #3: The authors developed an important clinical toxin study involving Hump-nosed pit viper accidents in Sri Lanka: evaluation of plasma and urinary markers for prediction of Acute kidney injury. The results are solid and have an excellent number of markers, all very well suited and directly related to the AKI event. The authors noted that Cystatin C was an important biomarker in predicting a severe/moderate event within 4 hours of poisoning, giving credibility to its use in the clinical approach. The results obtained in the present study are of great clinical importance for Hump-nosed pit viper envenomations, hoping for an improvement in the clinical management of patients.

As a major weakness of the study, the population size of moderate/severe AKI patients did not provided enough credibility to the study. Suggestion is to gather the severity groups in just AKI patients.

PLOS authors have the option to publish the peer review history of their article (what does this mean?). If published, this will include your full peer review and any attached files.

Reviewer #1: No

Reviewer #2: No

Reviewer #3: Yes: Marco A Sartim
---

## [Decision Letter · Decision Letter 1]

19 Nov 2021

Dear Prof. Wijewickrama,

We are pleased to inform you that your manuscript 'Serum and urinary biomarkers for early detection of acute kidney injury following Hypnale spp. envenoming' has been provisionally accepted for publication in PLOS Neglected Tropical Diseases.

Best regards,

Wuelton Marcelo Monteiro, Ph.D.

Deputy Editor

Wuelton Monteiro

Deputy Editor

Reviewer's Responses to Questions

**Key Review Criteria Required for Acceptance?**

**Methods**

-Are the objectives of the study clearly articulated with a clear testable hypothesis stated?

-Is the study design appropriate to address the stated objectives?

-Is the population clearly described and appropriate for the hypothesis being tested?

-Is the sample size sufficient to ensure adequate power to address the hypothesis being tested?

-Were correct statistical analysis used to support conclusions?

-Are there concerns about ethical or regulatory requirements being met?

Reviewer #1: Please see comments to the editor

Reviewer #2: (No Response)

Reviewer #3: Approved

**Results**

-Does the analysis presented match the analysis plan?

-Are the results clearly and completely presented?

-Are the figures (Tables, Images) of sufficient quality for clarity?

Reviewer #1: Please see comments to the editor

Reviewer #2: (No Response)

Reviewer #3: Approved

**Conclusions**

-Are the conclusions supported by the data presented?

-Are the limitations of analysis clearly described?

-Do the authors discuss how these data can be helpful to advance our understanding of the topic under study?

-Is public health relevance addressed?

Reviewer #1: Please see comments to the editor

Reviewer #2: (No Response)

Reviewer #3: Approved

**Editorial and Data Presentation Modifications?**

Reviewer #1: Please see comments to the editor

Reviewer #2: (No Response)

Reviewer #3: Approved

**Summary and General Comments**

Reviewer #1: Please see comments to the editor

Reviewer #2: (No Response)

Reviewer #3: Approved

PLOS authors have the option to publish the peer review history of their article (what does this mean?). If published, this will include your full peer review and any attached files.

Reviewer #1: No

Reviewer #2: No

Reviewer #3: No

---

## [Editor Report · Acceptance letter]

1 Dec 2021

Dear Prof. Wijewickrama,

We are delighted to inform you that your manuscript, "Serum and urinary biomarkers for early detection of acute kidney injury following Hypnale spp. envenoming," has been formally accepted for publication in PLOS Neglected Tropical Diseases.

Best regards,

Shaden Kamhawi

co-Editor-in-Chief

Paul Brindley

co-Editor-in-Chief
